# Full-Mouth Rehabilitation of a Patient with Sjogren’s Syndrome with Maxillary Titanium-Zirconia and Mandibular Monolithic Zirconia Implant Prostheses Fabricated with CAD/CAM Technology: A Clinical Report

**DOI:** 10.3390/jfb14040174

**Published:** 2023-03-23

**Authors:** Francisco X. Azpiazu-Flores, Damian J. Lee, Carlos A. Jurado, Kelvin I. Afrashtehfar, Abdulaziz Alhotan, Akimasa Tsujimoto

**Affiliations:** 1Department of Restorative Dentistry, Gerald Niznick College of Dentistry University of Manitoba, Winnipeg, MB R3E 0W3, Canada; 2Division of Restorative and Prosthetic Dentistry, College of Dentistry, The Ohio State University, Columbus, OH 43210, USA; 3Department of Prosthodontics, University of Iowa College of Dentistry, Iowa City, IA 52242, USA; 4Division of Restorative Dental Sciences, Clinical Sciences Department, Ajman University College of Dentistry, Ajman City 346, United Arab Emirates; 5Department of Reconstructive Dentistry and Gerodontology, School of Dental Medicine, University of Bern, 3010 Bern, Switzerland; 6Dental Health Department, King Saud University College of Applied Medical Sciences, Riyadh 1145, Saudi Arabia; 7Department of Operative Dentistry, University of Iowa College of Dentistry, Iowa City, IA 52242, USA; 8Department of General Dentisry, Creigthon University School of Dentisry, Omaha, NE 68102, USA

**Keywords:** full-mouth rehabilitation, CAD/CAM implant restorations, zirconia, implants

## Abstract

Dental implants have become a well-established treatment modality for the management of complete and partial edentulism. Recent advancements in dental implant systems and CAD/CAM technologies have revolutionized prosthodontic practice by allowing for the predictable, efficient, and faster management of complex dental scenarios. This clinical report describes the interdisciplinary management of a patient with Sjogren’s syndrome and terminal dentition. The patient was rehabilitated using dental implants and zirconia-based prostheses in the maxillary and mandibular arches. These prostheses were fabricated using a combination of CAD/CAM and analog techniques. The successful outcomes for the patient demonstrate the importance of appropriate use of biomaterials and the implementation of interdisciplinary collaboration in treating complex dental cases.

## 1. Introduction

Terminal dentition refers to a moment when the dentition is compromised to the point that its predictable restoration is no longer viable; this stage is usually reached after years of periodontal disease or as a result of unfavorable social or pharmacological conditions [1] or, in some situations, autoimmune diseases. Sjogren’s syndrome is an autoimmune condition characterized by lymphocyte infiltration and progressive destruction of the exocrine glands. One characteristic of the disease is a reduction in the production of tears and saliva, which has a negative effect on overall oral health and patient well-being and worsens over time. Dry mouth is the key diagnostic sign for this disease and can lead to a wide range of problems, including oral infections, progressive tooth decay, and periodontal breakdown [2]. When patients reach the terminal dentition stage, the restorative dentist faces the challenging task of devising a treatment that meets the patient’s esthetic and functional expectations. Traditionally, patients with several non-restorable abutment teeth had no choice but to use complete dentures, which, despite their long history of service, only restored the masticatory function to some extent [3,4]. Nowadays, thanks to dental implants, patients with terminal dentitions can be restored predictably and effectively with complete arch fixed prostheses.

Since the introduction of root-form dental implants by P.I. Branemark in 1965, dental implants have become a powerful resource for the rehabilitation of completely and partially edentulous patients [5]. Contemporary dental implants display high survival rates, which are the result of years of research directed at enhancing their topography, surface chemistry, and macroscopic features [6,7]. Thanks to these advances, contemporary dental implant protocols permit the rapid rehabilitation of complex clinical situations with a reduced number of dental implants [8,9,10]. These noteworthy advancements have also fostered the development of prosthetic materials, imaging and planning methods [11], surgical protocols [12], manufacturing technologies [13,14], and retention mechanisms [15] specifically created to optimize the restorative process.

Full arch reconstruction with implant therapy can be divided into screw-retained and cement-retained prostheses, and each option has shown advantages and disadvantages [15,16]. Screw-retained full-arch implant prostheses offer several advantages, such as retriability [17], clear access for hygiene maintenance of the prosthesis, implant, and surrounding tissues [18], and simple methods for repair [19]. Some disadvantages include the higher number of components and laboratory procedures, increased chairside time for the clinician, compromised esthetics, and screw loosening [20,21,22]. Cement-retained prostheses provide superior stability, esthetics, and occlusion in comparison to screw-retained prostheses [20,21,22,23]. However, some disadvantages include the difficulty of retrievability and the risk of having an excess of cement in the periodontal tissue [20,24].

Computer-aided design and computer-aided manufacturing (CAD/CAM) technologies make possible the creation of complex objects with minimal error [13,14]. These manufacturing technologies have been used extensively in engineering, medicine, and dentistry thanks to their versatility and wide range of applications. CAD/CAM technologies can be subtractive or additive, depending on how the object is created [13]. Additive manufacturing involves forming the object layer-by-layer [14], while in subtractive manufacturing techniques, the object is produced by cutting the material to the desired shape with a sharp cutting tool controlled by a computer [13]. Complex dental devices, such as surgical templates [25], dental models, custom trays, and dental prostheses, can be successfully manufactured with these manufacturing methods [12,14,26].

A great example of the advancements made possible thanks to CAD/CAM subtractive manufacturing are yttria-stabilized zirconia dental prostheses. Zirconia and other high-strength ceramics have been widely used in biomedical engineering and orthopedics for reconstructive purposes [27,28,29,30]. Zirconia is a polycrystalline ceramic that undergoes phase transformation when subjected to different temperatures. At temperatures greater than 2367 °C, zirconia has a cubic structure; between 1167 °C and 2367 °C, zirconia is tetragonal; and below 1167 °C, the structure is monoclinic [27]. This material is stabilized with dopants such as Mg, Ca, Sc, Y, or Nd to prevent its transformation from the high-strength tetragonal phase to the weaker monoclinic phase at room temperature [27]. With a flexural strength > 900 MPa and a high fracture toughness of 8 MPa·m^1/2^, stabilized tetragonal zirconia has gained enormous popularity as a dental restorative material [27]. In fact, research has demonstrated 5-year survival rates above 99% when the prosthesis is meticulously designed with pink felspathic porcelain limited to its gingival portion [31,32]. In addition to its remarkable mechanical properties, zirconia presents excellent biocompatibility, displaying no local or systemic cytotoxic effects [27,29] and reducing dental plaque accumulation when compared to other restorative materials [31]. These features have made stabilized tetragonal zirconia one of the most versatile and reliable contemporary restorative materials available. The present clinical report presents the comprehensive rehabilitation of a patient with Sjogren’s syndrome and terminal dentition using a maxillary titanium-zirconia complete arch prosthesis and a complete-arch mandibular monolithic zirconia prosthesis fabricated using a combination of analog techniques and CAD/CAM technologies.

## 2. Materials and Methods

A 69-year-old male patient presented to the Advanced Prosthodontics Dental Clinics at the Ohio State University seeking comprehensive dental care. At the time of the initial examination, the patient stated that he had medically controlled hypertension and Sjogren’s syndrome managed with salivary substitutes (Biotene Dry Mouth; GlaxoSmithKline Group, Durham, NC, USA). The extraoral examination revealed anterior metal-ceramic restorations, a positive smile line, and multiple missing anterior teeth (Figure 1).

Intraorally, all the remaining teeth except for the mandibular incisors had complete-coverage extracoronal restorations, the majority with secondary decay. A sinus tract was noticed on the buccal mucosa of tooth number 4.1, and tooth number 1.1 presented a horizontal fracture above the gingival margin (Figure 2). Additionally, clinical features typical of Sjogren’s syndrome, including minimal salivary flow and generalized bleeding on probing, were also noted during the examination.

The clinical findings were corroborated radiographically since multiple radiolucent lesions were noticed on the margins of the restorations, thus confirming the compromised state of the dentition. Additionally, a radiopaque mass was noticed on the anterior of the right mandibular angle (Figure 3).

After asking the patient, he explained that it was an osseous exostosis of benign origin that was frequently monitored by his primary care physician. Maxillary and mandibular preliminary impressions were taken with irreversible hydrocolloid (Geltrate; Dentsply Sirona North America, York, PA, USA) and were used to fabricate maxillary and mandibular diagnostic casts with type III dental stone (Buff Stone; Whip Mix Corp., Louisville, KY, USA). Additionally, a diagnostic cone-beam computerized tomography (CBCT) was taken at the end of the appointment.

After evaluating the information gathered during the clinical examination, multiple treatment options involving removable and fixed dental prostheses were presented to the patient. Once the treatment options were reviewed, the patient stated that he preferred not to undergo extensive restorative procedures to keep his teeth. He explained that fixed partial dentures never restored his smile nor his masticatory function since they developed decay and failed within a few months post-delivery. Removable partial dentures were not considered since the patient had had issues related to difficult maintenance, poor function, and discomfort with these prostheses in the past. After evaluating these factors and analyzing the patient’s expectations, complete-arch implant-supported prostheses were considered a feasible, definitive treatment capable of improving the patient’s overall quality of life in addition to restoring esthetics and function. The advantages and limitations of complete arch dental prostheses supported by dental implants were discussed. After the treatment duration (including the number of appointments), finances, and expectations were discussed, the patient decided to proceed with a treatment plan consisting of 4 maxillary and 4 mandibular implants with complete-arch Zirconia-based prostheses.

Maxillary and mandibular diagnostic teeth arrangements were fabricated with denture teeth (Blue-Line; Ivoclar Vivadent Schaan, Liechtenstein, Switzerland) and visible-light cure (VLC) denture bases (Triad VLC Denture Base Material; Dentsply Sirona North America, York, PA, USA) to establish the desired incisal edge position and future occlusal plane (Figure 4A). Additionally, the distance from the incisal edges to the lower border of the upper lip during a maximum smile was recorded and inscribed on the diagnostic casts to plan the position of the maxillary implants so that they would be 4 mm above the upper lip during function and smiling to ensure the concealment of the future prosthesis-tissue junction [33,34]. Subsequently, the diagnostic casts and diagnostic artificial teeth arrangements were scanned using a benchtop 3D scanner (E3 Scanner; 3Shape A/S, Copenhagen, Denmark) and were merged with the patient’s CBCT data in an implant planning computer program (BlueSky Plan V4; Blue Sky Bio, Libertyville, IL, USA) (Figure 4B). The placement of four standard-diameter dental implants (Tapered Screw Vent 4.1 × 10; Zimmer Biomet, Parsippany, NJ, USA) for the maxillary arch and four standard-diameter dental implants (Tapered Screw Vent 4.1 × 1 1.5 and 4.1 × 10 mm; Zimmer Biomet, Parsippany, NJ, USA) for the mandibular arch was planned digitally (Figure 4C), and a surgical template was designed in the same computer program. Subsequently, maxillary and mandibular bone reduction guides and bone-supported surgical templates were manufactured using a clear photopolymer (Surgical Guide V2; FormLabs, Somerville, MA, USA) in a stereolithographic (SLA) 3D printer (Form2; FormLabs, Somerville, MA, USA). Additionally, maxillary and mandibular interim complete dentures were fabricated by compression molding using heat polymerized polymethylmethacrylate (PMMA) denture base resin (Lucitone 199; Dentsply Sirona North America, York, PA, USA). On the day of the surgery, the dental implants were placed uneventfully using the computer-generated surgical templates, and the mandibular prosthesis was immediately loaded (Figure 4D).

After 4 months of healing, the maxillary implants were uncovered, and tapered abutments were installed (straight, 15°, and 30° tapered abutment systems; Zimmer Biomet Dental). Definitive impressions were taken using custom impression trays (Triad Tru Tray; Dentsply Sirona North America, York, PA, USA) and medium-bodied polyether impression material (Impregum Penta; 3M America, Saint Paul, MN, USA), and maxillary and mandibular definitive casts were fabricated with low-expansion type IV dental stone (New Fuji Rock IMP; GC America Inc., St. Alsip, IL, USA). Subsequently, verification devices were fabricated using low-shrinkage PMMA resin (Pattern Resin LS; GC America Inc., St. Alsip, IL, USA) and were used to verify the accuracy of the definitive casts (Figure 5A). During the same appointment, occlusion rims were fabricated to record the maxillomandibular relationships, aided by the verification devices [35] (Figure 5B–D).

The definitive casts were articulated in a semi-adjustable articulator (Denar Omni-Track; Whip Mix Corp., Louisville, KY, USA), and maxillary and mandibular artificial tooth arrangements were fabricated (Figure 6).

The artificial tooth arrangements were tried intraorally, and their esthetics, phonetics, vertical dimension, and centric relation were evaluated and deemed satisfactory (Figure 7).

The contours and occlusal relationships of the maxillary and mandibular tooth arrangements were used as blueprints for the manufacture of the definitive maxillary complete-arch implant-supported prostheses. The maxillary prosthesis consisted of a titanium framework with a cementable 3 mm-thick zirconia overlay (AccuFrame 360; Cagenix Inc., Memphis, TN, USA) (Figure 8A,B), and the mandibular prosthesis was a monolithic zirconia prosthesis (BarZero; Cagenix Inc., Memphis, TN, USA). It is worth mentioning that, preceding the fabrication of the definitive prostheses, printed prototypes were ordered and tried intraorally to refine the occlusion and esthetics and verify the centric relation (Figure 8C). At this stage, a custom incisal guide table was manufactured to ensure the accurate reproduction of the anterior guidance established with the prototypes in the definitive prostheses [36] (Figure 8D).

Subsequently, the definitive maxillary and mandibular prostheses were fabricated (Figure 9) and tried intraorally.

The passivity and accuracy of the fit were assessed clinically and radiographically (Figure 10), and the abutment screws were tightened to the recommended manufacturer’s recommended torque values.

Vertical dimension, centric occlusion, protrusive, and laterotrusive excursive movements were assessed and refined (Figure 11). Additionally, cleanability, esthetics, and comfort were deemed adequate by the patient (Figure 11).

Subsequently, home maintenance instructions and interdental brushes (ProxaBrush Go-Betweens Wide; GUM Sunstar America, Schaumbaum, IL, USA) were provided, and a hygiene program consisting of recall appointments every 6 months was established on the day of delivery. At the subsequent appointments, overall hygiene was reassessed and deemed adequate. Additionally, during these appointments, the patient expressed satisfaction with the function, esthetics, and confidence provided by the definitive prostheses (Figure 12).

## 3. Results

In the present clinical report, complete-arch implant-supported prostheses permitted the predictable rehabilitation of a patient with severely decayed terminal dentition. CAD/CAM technology is a valuable resource for restorative dentists since it enhances communication between the different members of the restorative team and permits the fabrication of complex prosthetic designs that maximize functionality and retrievability while minimizing complications.

## 4. Discussion

In the present clinical report, a patient with Sjogren’s syndrome and terminal dentition was predictably rehabilitated with complete-arch implant-supported prostheses composed of two functional biomaterials: titanium and zirconia. For decades, titanium has been the material of choice for dental implants due to its availability, machinability, biocompatibility, and favorable elastic modulus [27]. Similarly, dental professionals have embraced high-strength polycrystalline zirconia ceramics as restorative materials since they allow the consistent manufacture of highly-esthetic, tooth-colored restorations with remarkable flexural strength [27,28] and biocompatibility [27,29]. Nowadays, thanks to the advances in CAD/CAM technologies, the best features of these completely different materials can be merged into a single prosthetic design, and multi-material complete-arch implant-supported prostheses can be designed and manufactured digitally [13]. However, as with any other dental prosthesis, the success and adequate function of these complex prostheses lie in an adequate design and careful clinical refinement.

Since fixed complete dentures were introduced as a treatment for edentulism, significant changes have been made to the design of complete-arch implant-supported prostheses. For zirconia-based rehabilitations, limiting the application of porcelain to the gingival region, reducing the extension of cantilevers, and ensuring adequate prosthetic space have been advised to prevent biological and mechanical complications [31,32,37]. In the present clinical report, the maxillary zirconia overlay accurately copied the occlusal relationships and anterior guidance established intraorally with the prototypes and was supported by a titanium bar with 25 mm^2^ of cross-sectional area. The decision to use a prosthesis composed of two different materials was based on the presence of bilateral 10 mm-long distal cantilevers, which could create unfavorable flexural stresses in the ceramic overlay [27], although there is no clear evidence indicating a detrimental effect of distal cantilevers when their extension is small [37,38]. This design was selected over a monolithic alternative for the maxillary arch since it would permit retrieving and replacing the overlay portion of the prostheses if any complications occurred. On the other hand, for the mandibular definitive prostheses, the implant distribution and prosthetic space available permitted designing a prosthesis of dimensions that permitted adequate esthetics and function without compromising structural durability; therefore, a completely monolithic design was used.

Regardless of the significant progress in contemporary biomaterials and CAD/CAM technologies, there are aspects of prosthetic and implant dentistry that need further consideration. Substantial research has been done evaluating several restorative alternatives for complete-arch implant-supported restorations, and aspects such as retrievability, passivity, and occlusion have been researched [18,19,20]. Recently, alternative retention mechanisms and prosthetic designs involving high-performance polymers and novel ceramic-reinforced materials have been implemented to rehabilitate patients with complex dental needs and craniofacial conditions [16,39,40]. In the present clinical report, the combination of CAD/CAM and contemporary biomaterials permitted restoring the confidence, esthetics, and function of a patient exhausted by failing restorations. Research suggests that dental implants are a feasible modality to rehabilitate patients with terminal dentition caused by Sjogren’s syndrome. Satisfactory survival rates, low marginal bone loss, and biological complications comparable to those in healthy patients have been reported in the literature [41]. A systematic review of the topic suggests that, to ensure optimum outcomes, a hygienic prosthetic design and regular maintenance regime should be established on the day of delivery [40]. With satisfactory maintenance, implant-supported rehabilitations in patients with Sjogren’s syndrome can perform satisfactorily for many years, with case reports describing up to 13 years of service available in the literature [42,43]. In a similar way, a cohort study reported high satisfaction levels for this treatment modality, and 97% of the candidates would recommend dental implants to other patients with Sjogren’s syndrome [44,45]. However, it is worth noting that the majority of research available describes the clinical performance of traditional prosthetic designs such as traditional fixed complete dentures or porcelain-fused to metal complete arch rehabilitations. Therefore, research on the clinical performance of newer prosthetic designs and functional biomaterials in patients with Sjogren’s syndrome is needed.

Finally, this clinical report presents limitations related to the lack of cytologic, morphometric, and prospective clinical evaluation of the interaction of the biomaterials used with the tissues of the patient. Prosthetic factors such as passivity, restitution of phonetics, reestablishment of occlusion, and anterior guidance were assessed clinically throughout the different stages of the treatment, and osseointegration was evaluated radiographically. The lack of quantitative analysis is a common limitation of clinical reports, where time and patient factors play an important role. For these reasons, the materials used in the prostheses were not examined in greater depth in this clinical report. However, since their introduction as restorative materials, zirconia and titanium have demonstrated their biomechanical adequacy for complete arch rehabilitations when masticatory dynamics are considered [27,31,32]. Therefore, although the present clinical report lacks quantitative analysis, the gratitude of the patient and his favorable adaptation to the prostheses suggest the achievement of functionality and satisfaction, two of the greatest indicators of success in any restorative treatment.

## 5. Conclusions

A patient with Sjogren’s syndrome and terminal dentition was successfully rehabilitated using complete-arch implant-supported prostheses and dental implants. Complete-arch implant-supported rehabilitations manufactured using CAD/CAM technologies can be designed with multiple components made of different materials to ensure the prostheses have satisfactory biomechanics, esthetics, and occlusion.

## Figures and Tables

**Figure 1 jfb-14-00174-f001:**
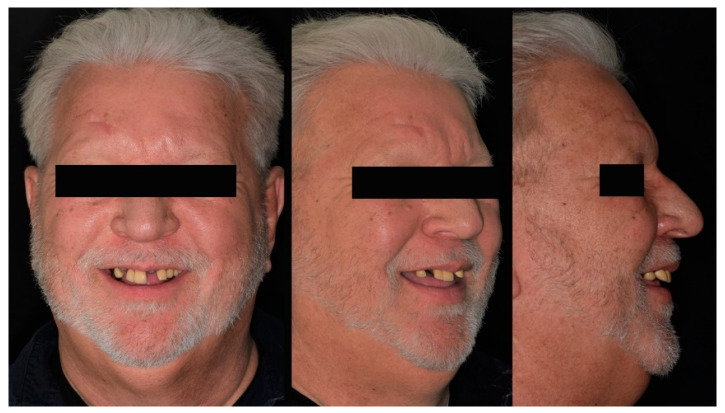
Pre-treatment extraoral photographs (from left to right) Frontal, ¾ profile, Profile.

**Figure 2 jfb-14-00174-f002:**
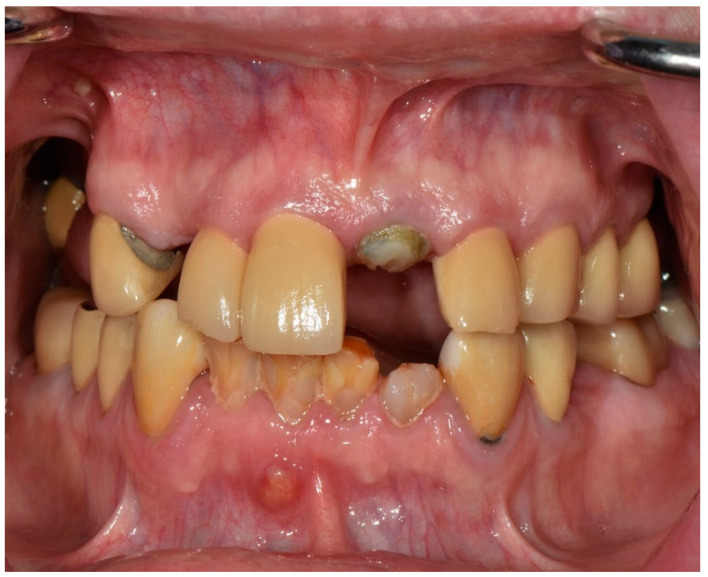
Frontal intraoral photograph taken at the initial appointment.

**Figure 3 jfb-14-00174-f003:**
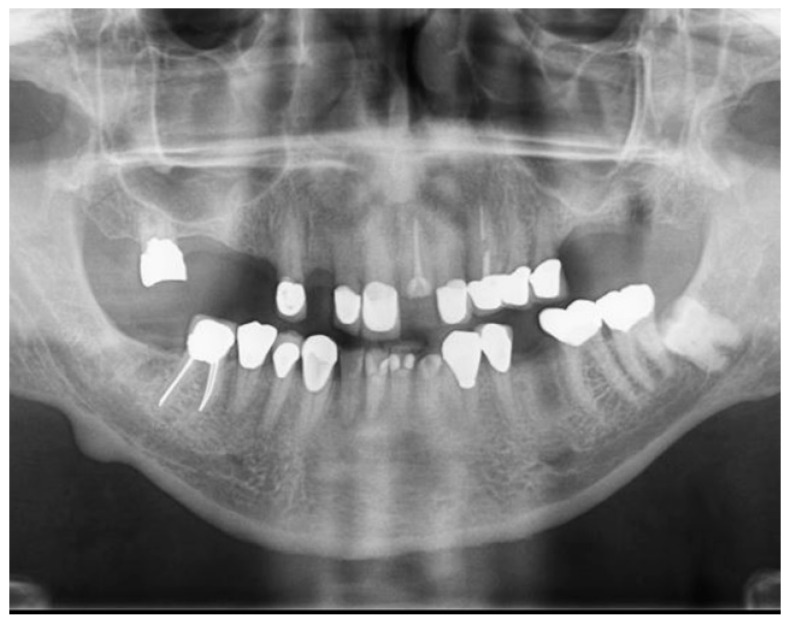
Pre-treatment panoramic radiograph.

**Figure 4 jfb-14-00174-f004:**
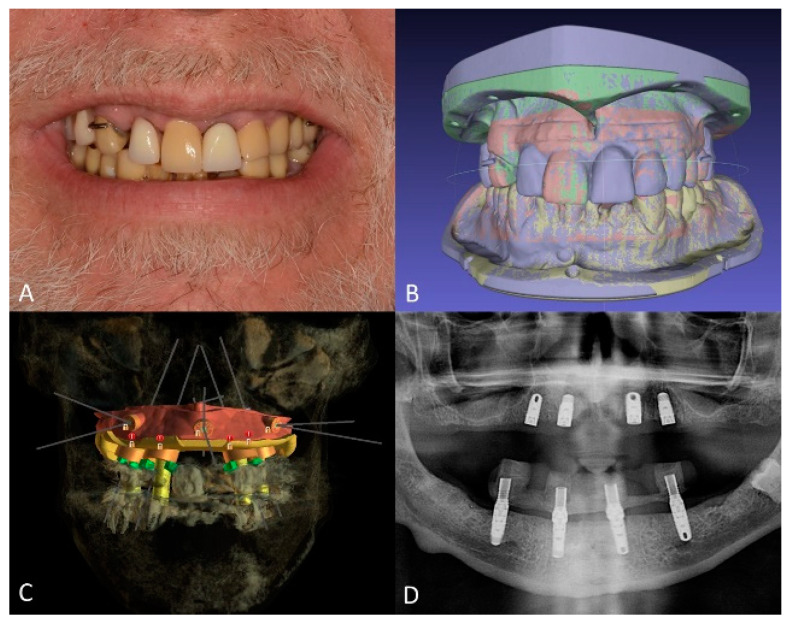
Treatment planning. (**A**), diagnostic artificial teeth arrangement. (**B**), 3D diagnostic models aligned in 3D modeling software. (**C**), planned dental implant positions, and surgical guide design. (**D**), panoramic radiograph of maxillary and mandibular dental implants immediately after placement.

**Figure 5 jfb-14-00174-f005:**
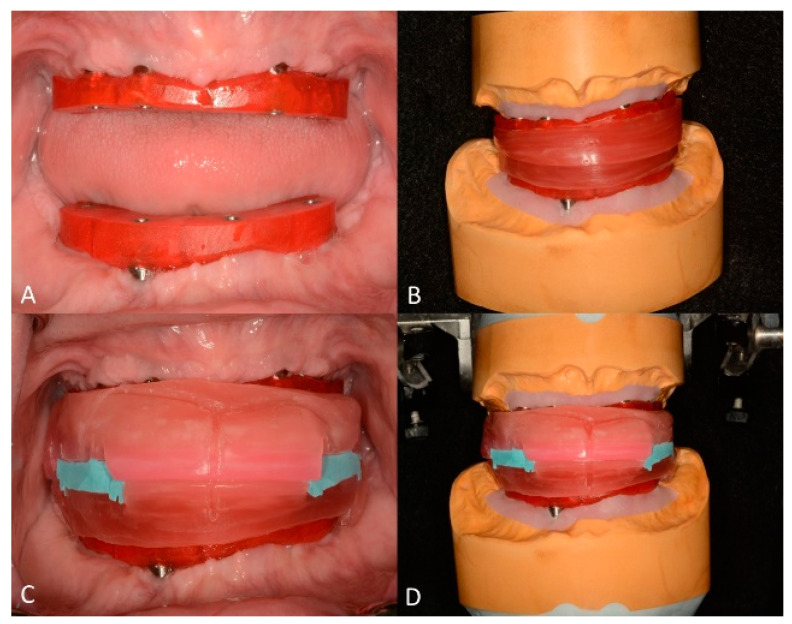
Definitive cast verification and maxillomandibular relationships. (**A**) intraoral photograph of maxillary and mandibular verification devices. (**B**) detachable occlusion rims. (**C**) intraoral photograph of maxillomandibular records (reprinted from: Azpiazu-Flores FX, Mata-Mata SJ. Overlay occlusion rim technique to facilitate the recording of maxillomandibular relationships. The Journal of prosthetic dentistry 2021;126:715-7 with permission from Elsevier [35]). (**D**) maxillary and mandibular definitive casts were articulated using the detachable occlusion rims supported by the verification devices.

**Figure 6 jfb-14-00174-f006:**
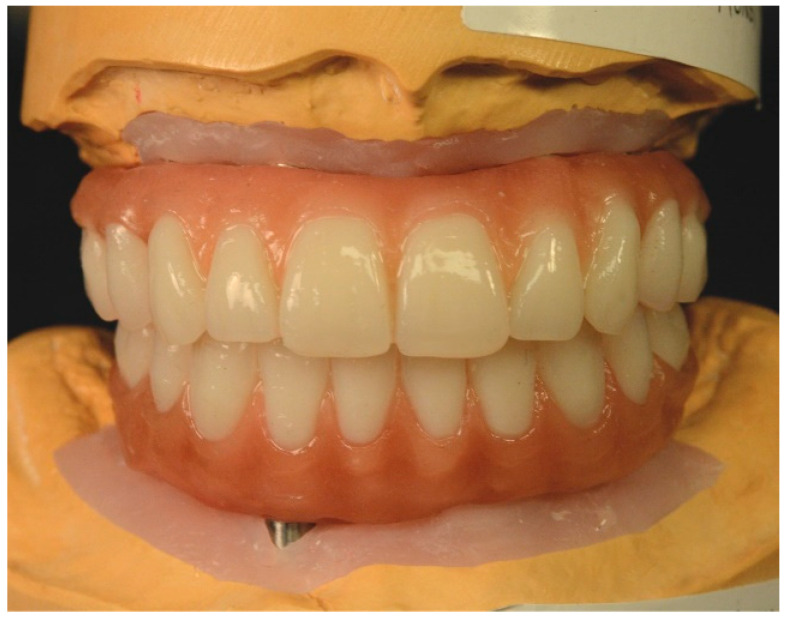
Artificial teeth arrangement to be used as blueprint for the definitive complete-arch implant-supported prostheses.

**Figure 7 jfb-14-00174-f007:**
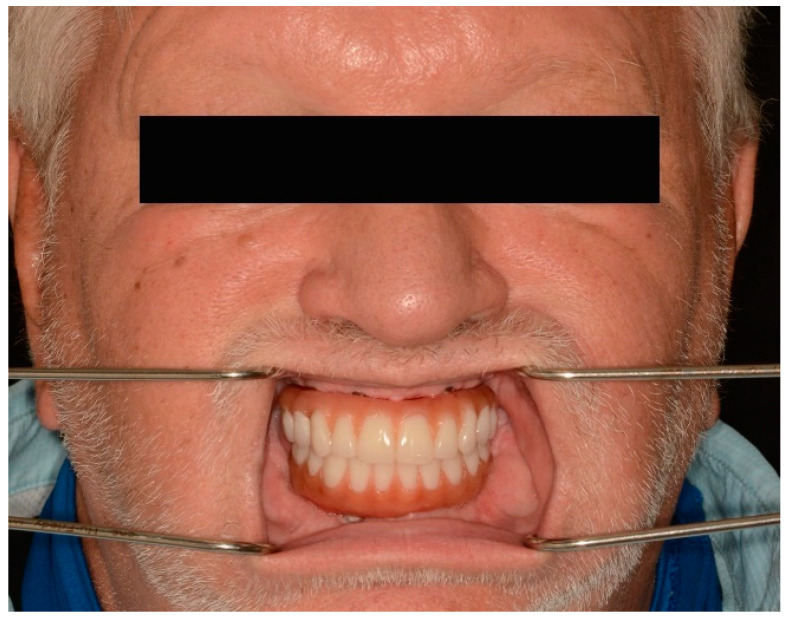
Extraoral photograph with retractors of artificial teeth arrangements.

**Figure 8 jfb-14-00174-f008:**
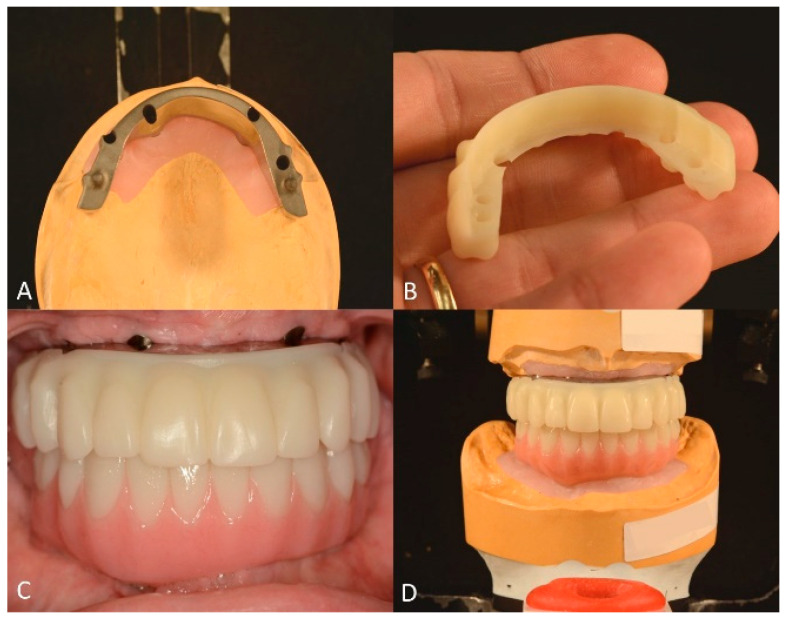
Maxillary and mandibular prototypes and titanium bar. (**A**), Occlusal view of CAD/CAM maxillary titanium bar. (**B**), maxillary PMMA overlay. (**C**), frontal view of maxillary titanium bar with PMMA overlay and mandibular prototype (**D**), and maxillary and mandibular prototypes mounted after intraoral adjustments with custom incisal guide table.

**Figure 9 jfb-14-00174-f009:**
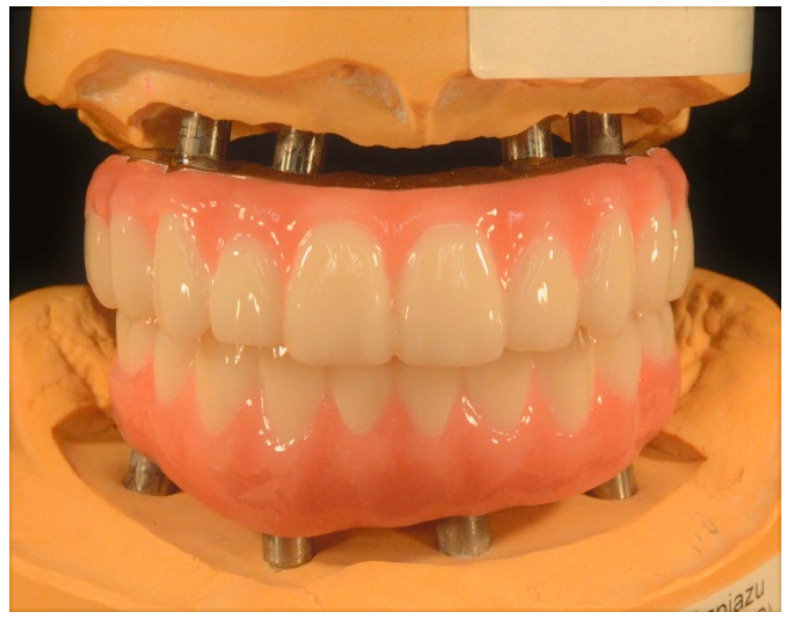
Maxillary and mandibular definitive complete-arch implant-supported prostheses.

**Figure 10 jfb-14-00174-f010:**
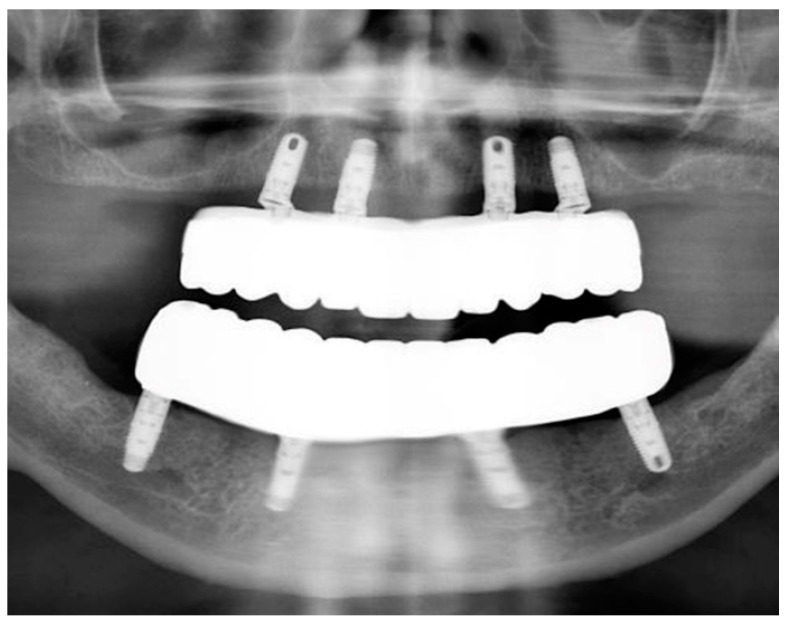
Panoramic radiograph of maxillary and mandibular prostheses taken at the time of delivery.

**Figure 11 jfb-14-00174-f011:**
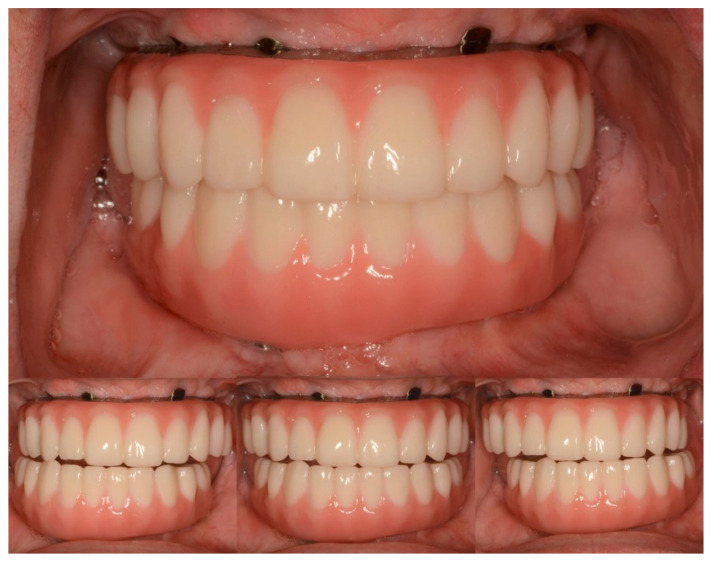
Intraoral photograph of maxillary and mandibular definitive complete-arch implant-supported prostheses. (**Top**) Centric occlusion. (**Bottom**) Lateral and protrusive excursions.

**Figure 12 jfb-14-00174-f012:**
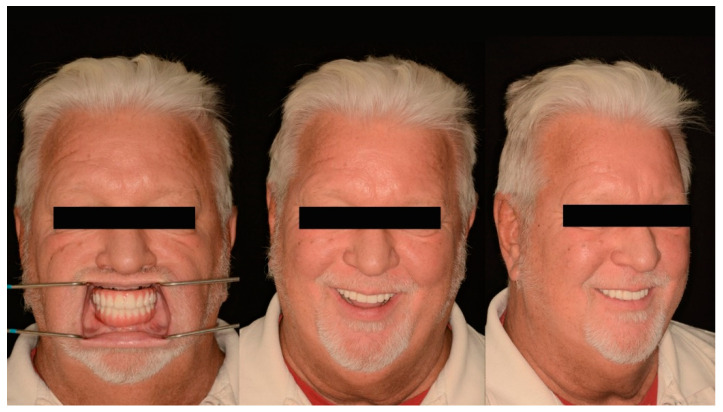
Post-treatment extraoral photographs (from left to right) Frontal with retractors, Frontal smiling, ¾ profile smiling.

## Data Availability

Not applicable.

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
