# Peer review of "Full-Mouth Rehabilitation of a Patient with Sjogren’s Syndrome with Maxillary Titanium-Zirconia and Mandibular Monolithic Zirconia Implant Prostheses Fabricated with CAD/CAM Technology: A Clinical Report"

_jfb, 2023, doi:10.3390/jfb14040174_

Round 1
Reviewer 1 Report
The article focuses on rehabilitation of the whole mouth with maxillary titanium-zirconia
and mandibular dentures on monolithic zirconia implants using
CAD/CAM technology
1. please provide histological studies of the tissue after implantation. in-vivo on animals, were the studies conducted?
2. how do new teeth affect cytology studies, what cultures grow in the mouth?
10.1016/j.ceramint.2019.04.081
3. Was morphometry performed?
10.1016/j.powtec.2020.04.040
4. Is the body's response present?
10.1016/j.pnsc.2019.07.004
5. The composition of the material from which the prosthesis is made is not presented, please give a brief description with the methods of research
Reviewer 2 Report
The article presents one very applicative method on how maxillary implant prostheses made of Titanium-Zirconia 2 and Mandibular Monolithic Zirconia can be realized using CAD/CAM Technology. The methods are quite well described, images presented are valuable and emphasize very well the key steps followed in the research. Methods that were utilized are also clearly explained in the text. Only few minor things I want to recommended to the authors / can be improved / I would like to suggest to the authors, such as the following ones:
1. Section about the Results is quite general and quite short as it is presented now. Maybe on this section could be added more things in detail so this section would have one page or so, or this section could be joined in as single section with the one related to Discussion.
2. In line 236 probably the source number should be [20] not [10] to be mentioned, since at is now source number [20] is not specified anywhere in the text of the article
3. Conclusions section is too short in my opinion and it looks like a repetition / a general presentation about what was done. In my opinion the Conclusions section must be little bit much longer as it now and will have to be focused on more concrete results that were achieved finaly (with quantitative indicators). Also one -two sentences about what can be still done in the future in the field could be added in this section. I think that there are still things that can be improved in the field, isn't it? Or it is not the case - it is nothing to be added / continued / improved in the field in the future?
4. Out of 21 reference sources, just three are dated in the period 2021-2022. I personally recommend the inclusion of at least 5-7 new references that are much closer to nowadays, besides the three ones already specified as being dated in the period 2021-2022. In this way one may have the feeling that the approached topic is relevant not for what was done 10 years ago, but it is in trend with the researches performed in the field in this period of time nowadays in the time when we are living (2022 - 2023).
Based on these observations, my decision is that the article can be accepted for publication after the minor revisions to be done by the authors as they have been suggested / provided above.
Reviewer 3 Report
The case presented is one of severely erroneous medical practice in which a large number of teeth are extracted without associated pathology, with good periodontal support and overall excellent prognosis, which would have allowed a multitude of durable and economically convenient prosthetic solutions.
There is, in my opinion, absolutely no justification to extract such a large number of practically healthy teeth, to be replaced by a small number of implants. I do not think that such a case can be published in a serious scientific journal
Reviewer 4 Report
The present case report described a Full-Mouth Rehabilitation with Maxillary Titanium-Zirconia and Mandibular Monolithic Zirconia Implant Prostheses Using CAD/CAM Technology. Although not extremely original, the manuscript case is interesting. A follow-up would be of interest.
The manuscript is well-written and readable.
The introduction, as well as the Discussion sections, are well organized.
The case reported is clearly described.
Please, add the definition case report to the title.
Round 2
Reviewer 1 Report
the article can be accepted in its current form
Author Response
Thank you.
Reviewer 3 Report
Thank you for your answer, which I still consider unsatisfactory for the following reasons:
1. General reasons. The main purpose of the dentist is to preserve dental tissues and in this case we have numerous vital teeth with excellent periodontal implantation and multiple conservative therapeutic options
2. Local reasons - the diagnosis of secondary caries justified only on the basis of panoramic radiography is questionable, and even in this case, on vital teeth with periodontal implantation and thick periodontal biotype there are numerous conservative options.
I strongly believe that in this case, good prognosis teeth should not be extracted to be replaced by complete arch restorations supported by only 4 implants
Author Response
Thank you for your comments. We have added further details about the patient's underlying conditions which justify the choice of treatment.